# New Landslide Disaster Monitoring System: Case Study of Pingding Village

**Yao Min Fang [1], Tien Yin Chou [1,*] , Thanh Van Hoang [1] , Quang Thanh Bui [2] , Duy Ba Nguyen [3] and Quoc Huy Nguyen [2,4]**

[1]  Geographic Information Systems Research Center, Feng Chia University, Taichung 40724, Taiwan; frankfang@gis.tw (Y.M.F.); van@gis.tw (T.V.H.)
[2]  Faculty of Geography, VNU University of Sciences, 334 Nguyen Trai Str., Thanh Xuan, Hanoi 100000, Vietnam; qthanh.bui@gmail.com (Q.T.B.); st_huy@gis.tw (Q.H.N.)
[3]  Department of Photogrammetry and Remote Sensing, Hanoi University of Mining and Geology, No. 18 Vien Street—Duc Thang Ward—Bac Tu Liem District, Hanoi 100000, Vietnam; nguyenbaduy@humg.edu.vn
[4]  PhD program of Civil Engineering, Water Resources Engineering, and Infrastructure Engineering, College of Construction and Development, Feng Chia University, Taichung 40724, Taiwan
*   Correspondence: jimmy@gis.tw; Tel.: +886-42451-6669

**Abstract:** The Linbeiken area is located in the village of Pingding, Taiwan. Since the Mindulle and Aere Typhoons in 2004, and as a result of the landslide triggered by the continuous heavy rainfall on 9 June 2006, there has been a persistent collapse of side slopes in the area. This paper describes the equipment that was installed to collect on-site topographic and hydrological information in the Linbeiken area upstream of the Pingding River and to monitor changes in the landslide area, as well as the measurements that were collected during the 2008 Typhoon Sinlaku. A case study of a landslide in Pingding, Taiwan was used to monitor the accurate coordinate changes in the potential landslide areas during typhoons. The goal of this study was to establish warning indexes, and to strengthen the software and hardware at the local disaster response center in the hope of gaining a full idea of the surface movement in landslide areas in future flood seasons. This is important for boosting the preparedness to adapt to landslide hazards, for improving disaster warnings, and for reporting efficiently to better protect the lives and property of local residents. The results show that the landslide disaster monitoring and warning system in Taiwan, as applied during Typhoon Sinlaku in 2008, is both effective and comprehensive.

**Keywords:** Global Positioning System (GPS); landslide; disaster prevention monitoring; disaster mitigation

## 1. Introduction

Landslides cause significant damage to both people and property. Many studies have researched landslides, such as that of Chen et al. [1], which showed the application of the three-dimensional deterministic model to a landslide event in Taiwan. Hsu and Liu integrated the TRIGRS (Transient Rainfall Infiltration and Grid-Based Regional Slope-Stability) and DEBRIS-2D (debris flow-two dimensional) models for landslides in Taiwan [2], while Bunn et al. and Ramos-Bernal et al. [3,4] presented the LIDAR (Light Detection and Ranging) digital elevation model and ASTER (Advanced Spaceborne Thermal Emission and Reflection Radiometer) imagery research on landslides. Liu et al. [5] used a geographically weighted regression model for studying landslides in the QingChuan area of China. Liu et al. [6] used a variety indexes like the C-, X-, and L-band synthetic aperture radar

(SAR) datasets for landslide analysis in China. Assilzadeh et al. applied GIS application for landside prevention Penang Island in the Straits of Malacca [7], while Long et al. [8] analyzed the mapping of rainfall-induced landslides in Vietnam. Chou et al. applied unmanned aerial vehicle technology to produce a digital elevation model-based dataset with a 5 m resolution for disaster monitoring and management operations [9]. In recent decades, many different techniques have been developed that focus on early warning disaster monitoring and management systems. The study by Anita et al. [10] was based on using sensor nodes to analyze the data for disaster monitoring. Kwak analyzed multiple data sources of various satellite for disaster risk reduction [11]. McCarthy et al. created a GIS (Geographic Information System) expert system framework for the detection and monitoring of hazards [12]. Arattano et al. combined analytical sensors for the warning and monitoring of debris flow [13]. Xuan et al. provided vivid examples of and suggestions for improving monitoring systems [14]. Lucas et al. integrated characterization and monitoring [15]. Trocone et al. analyzed slope response by using the material point method [16].

With the Taiwan Strait on the left, connecting the island and Eurasia, and the Pacific Ocean on the right, Taiwan is located in one of the areas around the world that experiences the most monsoons. Prevailing winds change significantly as seasons alter; the rainy season and typhoons bring abundant rainfall and often cause landslides and landslips. Landslides are referred to as heterotaxy and occur instantly, while landslips are slow and are known as long-term heterotaxy. The Global Positioning System (GPS), which has been widely used in many academic subjects and fields, has proved to be a powerful tool for monitoring artificial and natural structure deformations, as well as slope displacements. GPS has a great number of advantages compared with traditional measurement technologies. Concisely, GPS is more precise, effective, and highly automatic, with low labor intensity required.

Wu analyzed a landslide event in Taiwan (during the 2009 Typhoon Morakot) by combining landslide susceptibility data and drawing a map [17]. In recent years, many different types of technologies have been developed, such as in case studies of Korea and Germany [18–20], which were focused on several technologies for disaster mitigation. Estrela et al. [21] used cameras to observe ground movement and reduce associated risks, while Zhao et al. researched SAR images for landslide monitoring [22]. An operational web-based GIS for the early warnings of landslides in Bangladesh, created by Ahmed et al. [23], is capable of providing alerts five days in advance. Other research works have resulted in the flooding model [24], the flooding forecasting system [25], and dynamic modeling for reservoir watersheds [26], which are all widely applied in disaster management.

On 25 June 2003, the 42–45 K section (Wuwanzi) of A Li Mountain Highway was affected by the most serious landslide in its history due to continuous torrential rain. With ceaseless rain-wash, the slumped section enlarged, and an area of nearly 120 m slipped off of the valley. As a result, the A Li Mountain Public Works Section commissioned Feng Chia University to install various transmission devices and equipment in the 31 K section of the Taiwan 18th Line Highway. As an outcome of this installation, it was suggested that reference values for rainfall levels, water levels of barrier lakes, drilled underground water levels of deposition areas, and overflow water levels be established and then used as alarms when exceeding such values.

Abnormal phenomena such as up-warp and crack-openings once again affected the dipped-slope site of the collapse zone in November 2002. To understand whether the Jiufenershan Collapse Zone was continuously sliding, the Soil and Water Conservation Bureau immediately organized the "Jiufenershan Collapse Zone Observation Project" to gain a primitive understanding of land surfaces. Observations of underground changes via geological drilling, land surface changes observation, the interior inclination of drilling holes, and underground observations can be used as references for emergency responses during typhoons or torrential rain [27].

The village of Pingding in the county of Yunlin, Taiwan was chosen as a study site. This village is located in a valley surrounded by steep mountains. The dynamic sediment process and the arrangement of the sensors are supposed to provide alerts six hours in advance of any potential landslides so as to

protect this village. The advantage of this study is its demonstration of a real-time monitoring station in Pingding that provides the residents with a landslide warning and management system. The aim of this research was: (1) To receive real-time data and to monitor the entire study area (high landslide potential area); (2) to integrate software and hardware for preparedness to adapt to landslide hazards; (3) to send warnings to residents to avoid serious landslides. The research procedure is indicated in Figure 1.

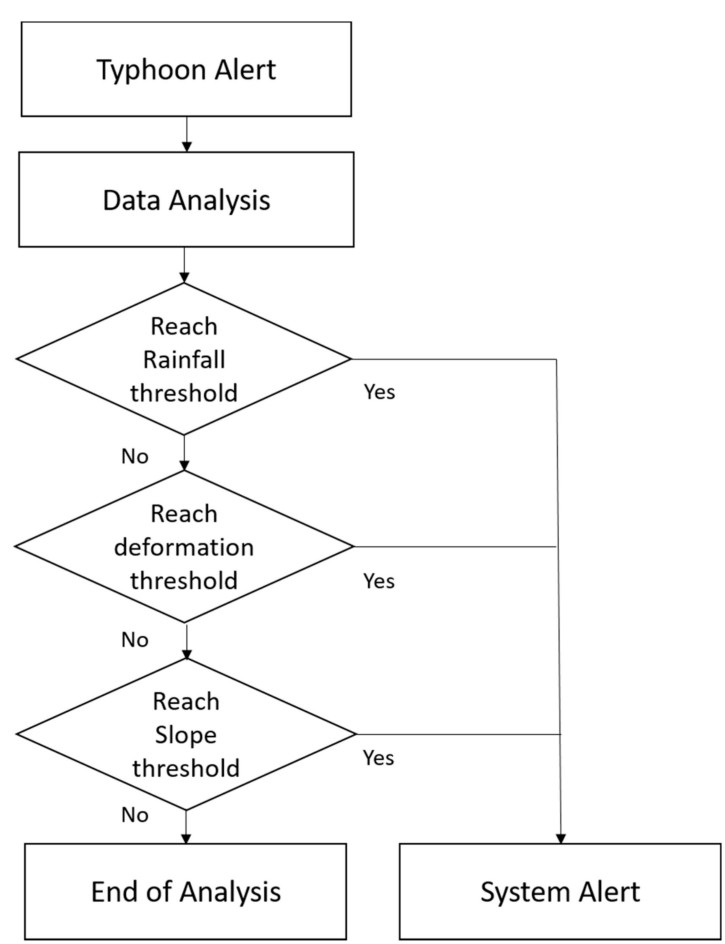

**Figure 1.** Flowchart of the research procedure.

## 2. Material and Methods

### 2.1. Study Area: The Village of Pingding

Pingding is situated in the township of Linnei in the county of Yunlin, Taiwan, and its geographical coordinates are 23°45′36″ N and 120°36′38″ E (Figure 1).

Linnei is divided into mountainous areas and plains by the Taiwan 3rd Line Highway. With mountains on its southeastern side and plains in the northwest, Linnei's altitude is 70–320 m above sea level, and its topography is a slope extending roughly from the southeast to the northwest. Pingding is situated on a mesa-type hill, with an altitude of 325 m above sea level. With a tropical and humid climate, the village has a yearly rainfall of 1500–2000 mm, which is concentrated in the summer. The exposed geological formation is the Toukoshan Formation Huoyanshan Conglomerate Section; the gravel components of the conglomerate are of less than 20 cm in size and are unevenly distributed. The sandstone and mudstone, which are sometimes mixed in the conglomerate, are mainly composed of sandstone and quartz sandstone. The exposed parts of the conglomerate often form cliffs or precipices and gorges if there are any valleys. The Pingding Collapse Zone is located within Pingding

in the mountains east of Linnei. Linnei, which is located in the northeast of Yunlin county (one of the counties in Taiwan), adjoins Pingding, Pingding Mountain, and the township of Zhushan, Nantou.

In terms of its geographical location, Pingding is located between Shexi and Gaoxikou in the township of Gaoshu (Figure 1). The study site is flat, tilting north and west, and the west is similar to a valley (Figure 2).

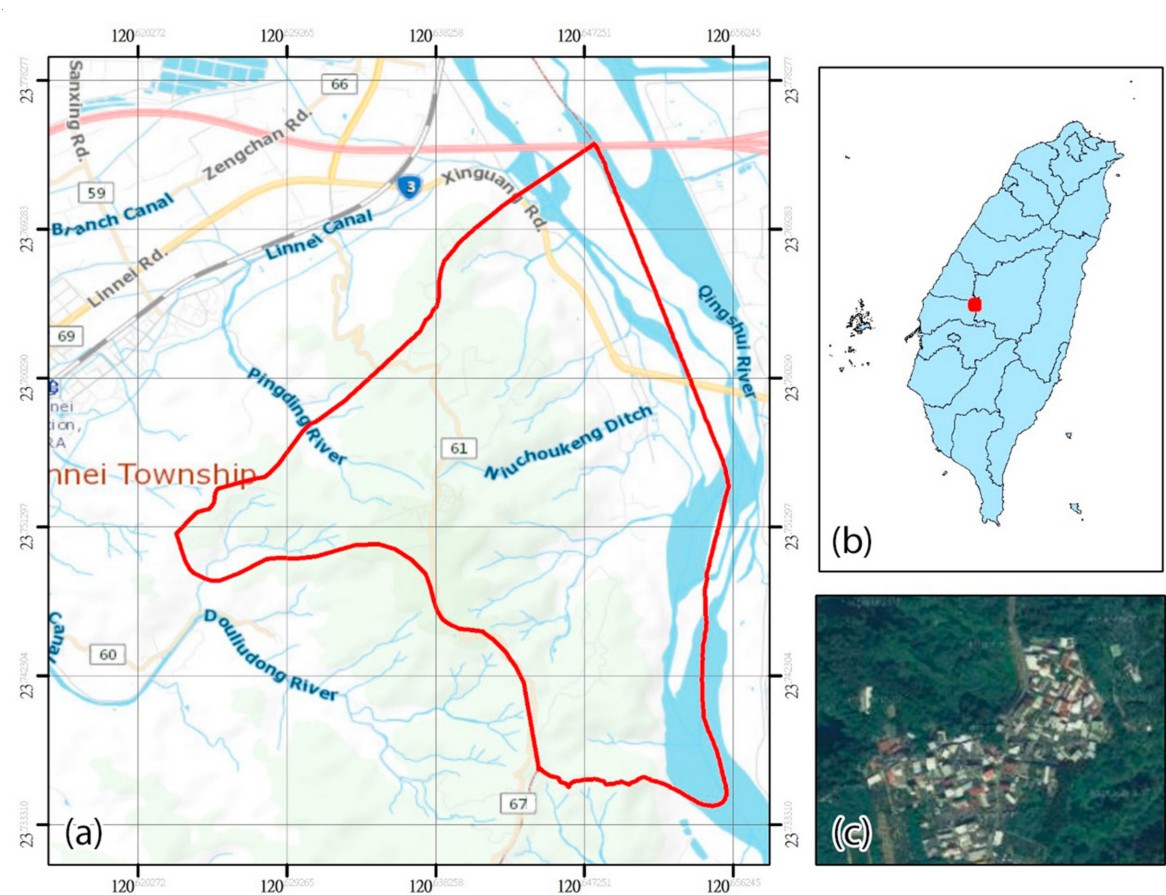

**Figure 2.** Pingding in Taiwan: (**a**) Pingding in the township of Linnei; (**b**) Pingding in Taiwan; (**c**) satellite image of Pingding.

According to the "Central Taiwan Geological Map—From Zhushan to Jiayi" document from the fourth issue of the Central Geological Survey Article Collection and Annual Report of Taiwan Council of Agriculture by the Soil and Water Conservation Bureau, the exposed strata near the Pingding Collapse Zone include the Huben Stratum and the Sanxing Stratum of the Pleistocene Age, as well as the Accretion of the Holocene. Judging from the positions and patterns of these strata, the Pingding Collapse Zone is located in the range of the Huben Stratum and the Sanxing Stratum, while both of the shores of the Pingding sub-stream are in accretion. According to the on-site geological surveys and the six (BH-1–BH-6) (geographical surveying site from BH-1 to BH-6) 12–30 m deep geological drilling holes, the strata can be further divided into reddish brown silty clay, brown sandy silty soil in the colluvial rocks, brown silty soil in the gravel stratum, and brown or grey siltstone.

### 2.2. Analysis of the Causes of Collapse in Pingding

The southwestern hills of Pingding belong to the Toukoshan Formation; the poor cementing of the hills' conglomerate and sandstone tissue raises the possibility of collapse due to washouts. As a result of the frequent headward erosion in the Linbeiken area (Pingding River), the platform has shrunk in the southeastern direction, leaving only the Pingding area behind. Enveloped in serious erosion, the entire

mountain area collapses easily when rainfall occurs. The proposed cause of this quick downward flow of the seepage water in the above soil stratum is because of the excellent permeability of the red soil and gravel strata on the surface. The seepage water, however, gets stuck here when confronted with gray sandy mud strata, which are poor in permeability and move horizontally. The underground water finally causes a collapse due to the decompression effect following seepage through the soil from the side slope surface. As the water content in the mud stratum increases, the mud stratum's ability decreases its shear resistance strength. These are key factors that contribute to the damage caused to the side of slopes, and can be judged from the locations and types of previous collapses. Generally, large-scale types of mass collapse are the main factors causing damage, while collapse locations are influenced by the load of slope tops, the water content of local soil (rock) strata, and the gradient.

## 2.3. Historical Disasters

The 921 Earthquake in 1999: The serious collapse areas that developed during the 921 Earthquake included the northeast slumping side slopes and the northwest Linbeiken Collapse Zone. The Linbeiken Collapse Pit experienced collapses of different scales, and despite many restoration works being done, the Linbeiken Collapse Pit has doubled in size in the past decade. The Yunlin County Government completed construction of the reinforced soil retaining wall in April 2003.

Typhoons Mindulle and Aere in 2004: Landslides once again struck the south bank of Linbeiken in September 2004 (Figure 3). The areas with reinforced soil retaining walls and part of the north-side tea garden collapsed into the Linbeiken Collapse Pit. The collapse created a stage terrain with a 30 m height difference. The collapse area of the first landslide covered approximately 1 ha. Afterward, it was followed by several collapses and slumps of different scales. The damage caused by this collapse included the slump of a 2 m$^2$ tea garden, but there were no casualties. Residents from four dwellings moved for safety purposes. Regarding public facilities, a diversion ditch of approximately 23 m in length and three areas of PVC (polyvinyl chloride) piping collapsed and were pulled apart.

Continuous torrential rain in June 2006: The flood on 9 June 2006 caused the collapse of the gabion slope protection on the north side of the Linbeiken source bank and the wash-away of part of the Yunlin 61st Line Highway roadbed.

22 May 2007: At 6 a.m., with no influence of rainfall, the original collapse area of the Yunlin 61st Line Highway suffered from another topple, damaging two-thirds of the main access road of the nearby villagers, making it impassable. Emergency work such as fence establishment and warning sign installations were carried out. The main reason for this collapse was the gravity effect influenced by saturation of the soft soil caused by the cracking of tap water pipes of the area, which led to the seepage of tap water and drizzle on previous days.

## 2.4. Data Collection

This section shows the organization of the research methods used. Figure 4 provides a flowchart of the research methods adopted to create a disaster monitoring and management system. First, for simulating the landslide disaster monitoring system, we visited the site to carry out surveys. Second, based on the results of the survey, we decided on and installed the appropriate equipment for the study site. Third, the thresholds for the sensors were defined. Fourth, based on the results of the analysis, we identified the accuracy of the measurements and validated the threshold value. Finally, a landslide monitoring displaying system in real time was constructed.

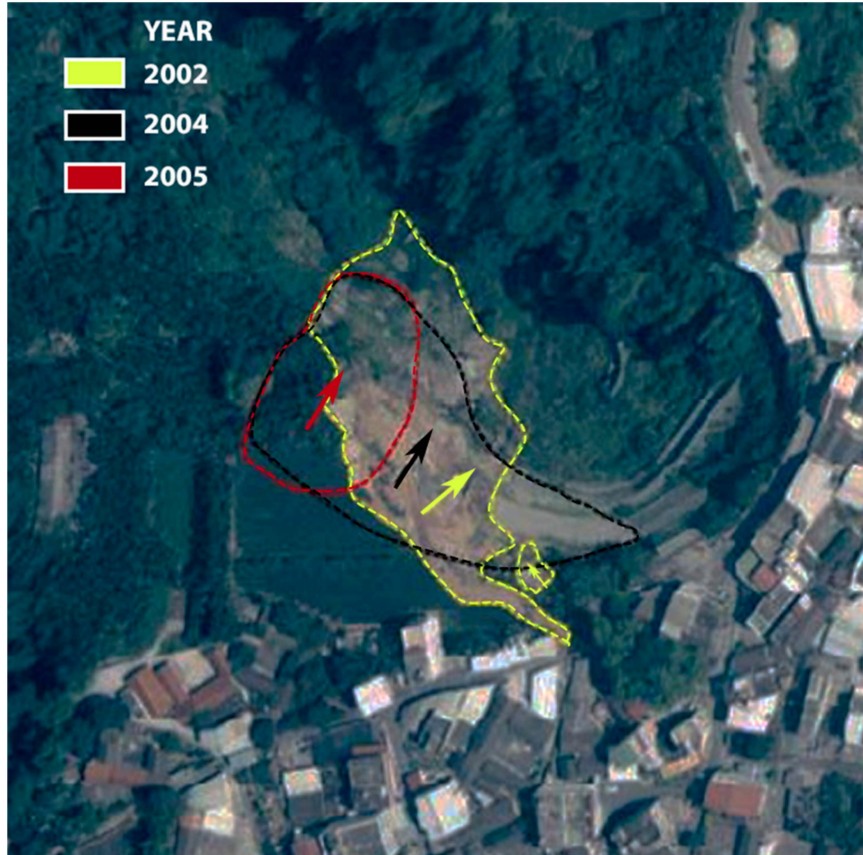

**Figure 3.** Collapse areas of Pingding River upstream, Linbeiken, and Pingding across different periods of time [28]. Note: The yellow parts refer to the collapse on June 2002 (caused by Jiji Earthquake and Typhoon Toraji); the black parts are the collapses on 9 September 2004, while the parts in red are the collapses on 11 January 2005. The arrows indicate the sliding directions of each collapse.

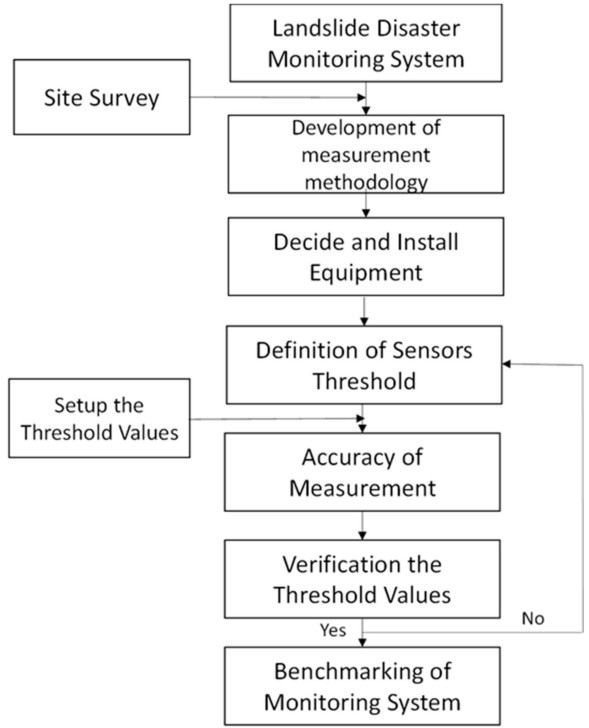

**Figure 4.** Research methodology flowchart.

2.4.1. Equipment for the Collapse Monitoring System

The Linbeiken area in Pingding, upstream of the Pingding River, has faced constant side slope collapses during torrential rain seasons since the 921 Earthquake in 1999. To assess the condition, this study monitored the collapse changes of the Pingding area and established a side slope collapse monitoring system to understand the real-time activities of the collapse zone during flooding (Figure 3). The following work was planned: The establishment of relevant monitoring equipment, including camcorders and devices and equipment related to rainfall monitoring and land surface deformation monitoring, as well as recording, storage, transmission, and display systems for monitoring the above equipment. In light of the simulation analysis results of Kun-yi Chen [29], the completed work included an inbuilt GPS-4 and GPS-5, a retractable cable meter (group 3), a retractable cable meter (group 4) on the sliding surfaces, an inbuilt retractable magnetic induction meter (groups 1), and a retractable meter (groups 1 and 2) in two open spaces that contained obvious cracks and may have suffered from possible continuous displacements. Finally, GPS-1, GPS-2, GPS-3, and slope circles were placed on the top of the fixed soil retaining wall. The details of the monitoring device numbers are shown in Table 1; the monitoring station is equipped with the following sensors: (1) Two sets of rain gauges; (2) three sets of CCD (A charge-coupled device) camcorders; (3) two sets of high-pressure illuminators; (4) one set of remote camcorders; (5) four sets of retractable meters; (6) one set of slop circles; (7) one sets of GPSs. The device arrangement is displayed in Table 1, Figure 5, which highlights the slope deformation monitoring system contained in the GPS receiver base station and five GPS receiver mobile stations. The precision of the real-time coordination solution was ±3 cm, while the solution speed was over one piece of information (including one) every 3 s.

**Table 1.** Quantities and descriptions of the observation facilities [30].

| Devices | Numbers | Brand | Device Functions |
|---|---|---|---|
| Rain gauge | Two sets | Taketa Keiki Industry Co., Ltd. Itabashi City, Tokyo 173-0024, Japan | Observing rainfall as a reference for real-time rainfall vigilance values |
| CCD (A charge-coupled device) camcorder | Three sets | AXIS Lund, Sweden | Observing local slopes and transmitting back real-time local images |
| High-pressure illuminator | Two sets | Micro Balance Wuqi District, Taichung City, Taiwan | Assisting the night-time observation of CCD camcorders |
| Remote camcorder | One set | AXIS Lund, Sweden | Helping to make the controls on the way to the Command Post remote to aid in the understanding of the on-site situation |
| Retractable meter (retractable displacement meter) | Four sets (10 m) | Novotechnik Ostfildern, Germany | Monitoring the changes in land surface cracks as a reference for emergency evacuation |
| (Dual direction) Slope circles | One set | Micronor Inc. Camarillo, CA 93012, United States | Measuring the changes in buildings' trace angles of inclination |
| GPS slope deformation monitoring system | One set (six GPS) | Leica Wetzlar, Germany | Monitoring the slope spatial 3D change condition through GPS as a reference for emergency evacuation |

GPS, Global Positioning System; 3D, three-dimensional.

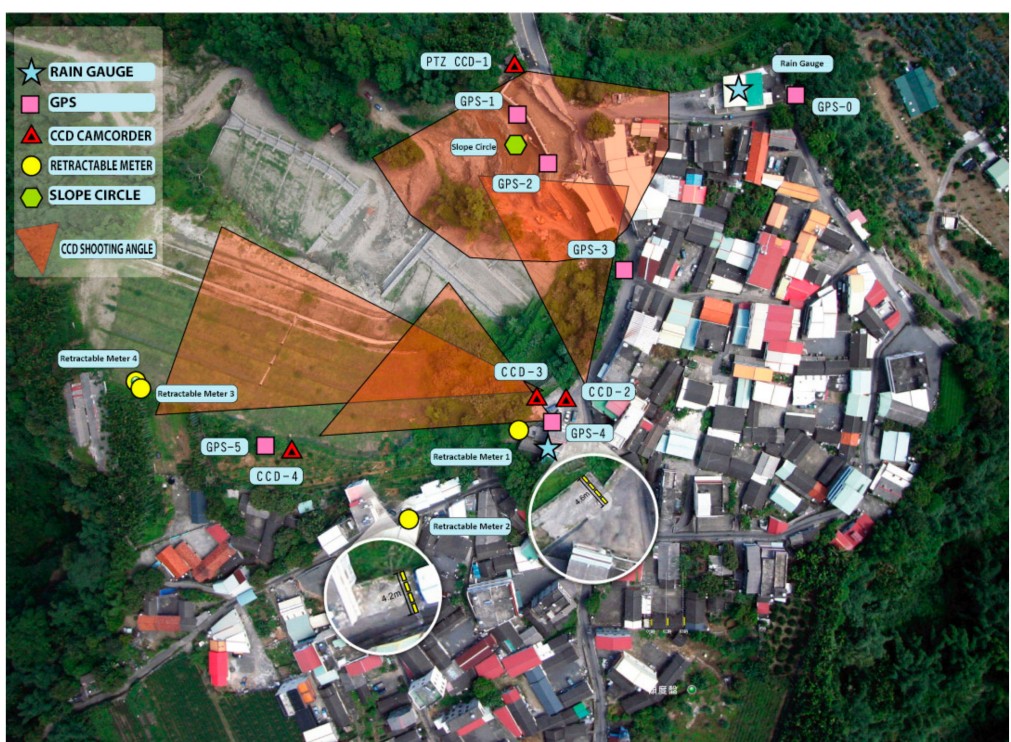

**Figure 5.** Arrangement of the devices presented in Pingding. The monitoring station is equipped with a rain gauge, a CCD camcorder, a high-pressure tiltmeter, a remote camcorder, a geophone, and slope circles (dual direction).

### 2.4.2. Establishment of Management Values

The fact that side slope failures often take place after rainstorms has led to the belief in a certain relationship between side slope failures and rainfall. Indeed, this relationship between side slope disasters and rainfall has been noticed and proven by researchers such as Brand [31], Lumb [32], and Slosson and Larson [33]. Nonetheless, some other prior hydrological conditions are essential for the triggering of side slope failures. Pre-storm rainfall injects water into side slope surfaces and allows water to flow freely in the slope. In other words, the soil surface needs to be saturated to trigger the following rainstorm's side slope failure mechanism. The pre-storm rainfall required by side slope failures depends on the soil surface cover, the water conduction capacity of the soil, the seepage rate, evapotranspiration, and the hydrological condition of the side slope.

Knowing that the influences of pre-storm rainfall on side slope stability have been studied for years, Lumb [32] discovered the impact that pre-storm rainfall has on side slope failures. In particular, he found out that higher pre-storm rainfall leads to worse side slope failures. He categorized side slope failures caused by rainfall into the following four groups:

- Severe events that cause over 50 side slope failures per day;
- Serious events that cause 10–50 side slope failures per day;
- Minor events that cause less than 10 side slope failures per day;
- Independent events that cause only one side slope failure per day.

## 3. Results

### 3.1. Monitoring Warning Systems

The data of different devices can be displayed in the following ways: Image data displays, cover real-time image displays, and historical image displays, as demonstrated by Figure 6.

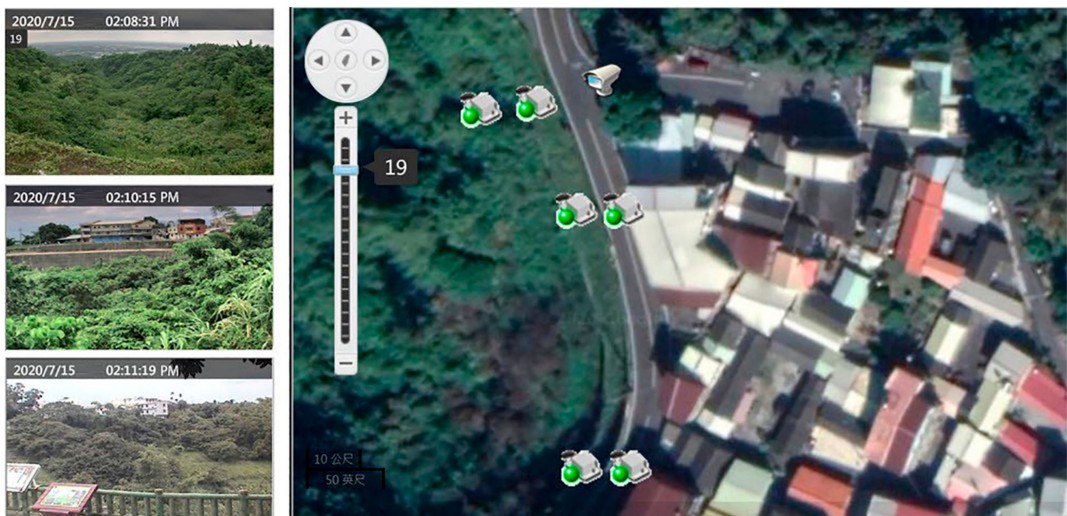

**Figure 6.** Real-time image displays of the status of the study site (green dots are normal status).

Two types of real-time image displays are available: Single-station displays and loop displays of data from all of the observation stations. Historical image displays allow users to browse the CCD images of certain chosen time periods through the single-station display mode. The slope displacement condition is shown through the displays and analyses regarding the space transformation condition of the retractable meters (tensile strain meters) and the GPS slope land deformation monitoring systems. Users are allowed to access the displacement condition through direct clicks on the devices shown in the device arrangement figure, which is a vigilance data reference provided for involved staff, as displayed in Figure 7. There are four groups of retractable meters, while the data displays show the conditions of the devices using red, yellow, and green lights, as per the display mode of the device set up procedures. When the displacement quantities of the retractable meters exceed the set threshold of the vigilance values, the system will switch the light to red or yellow to remind the responsible personnel of the meters' current condition. The GPS slope land deformation monitoring systems also display the devices' condition through red, yellow, and green lights, and the lights are switched to red or yellow as a reminder for personnel responsible for the current GPS conditions, as shown in Figure 8. In both situations, users are also permitted to click on the red or yellow light icons to inquire about the detailed displacement condition of a particular piece of equipment. The slop circle data displays analyze and display data, focusing on the condition of the angle changes of the slop circle. Users are allowed to access the displacement condition through direct clicks on the devices shown in the device arrangement figure, which is a vigilance data reference provided for relevant staff, as displayed in Figure 9. The real-time monitoring data of all of the devices and the arrangement figures of the relevant devices are shown in Figure 10.

### 3.2. Monitoring Data Analysis

During Typhoon Sinlaku, the emergency response team of the Soil and Water Conservation Bureau was active from 08:40 on 11 September 2008 to 18:32 on 19 September 2008. It is a hyetograph of the total cumulative rainfall of the entire island, while the cumulative rainfall of the Yulin area, which reached over 300 mm, is demonstrated in Figure 11.

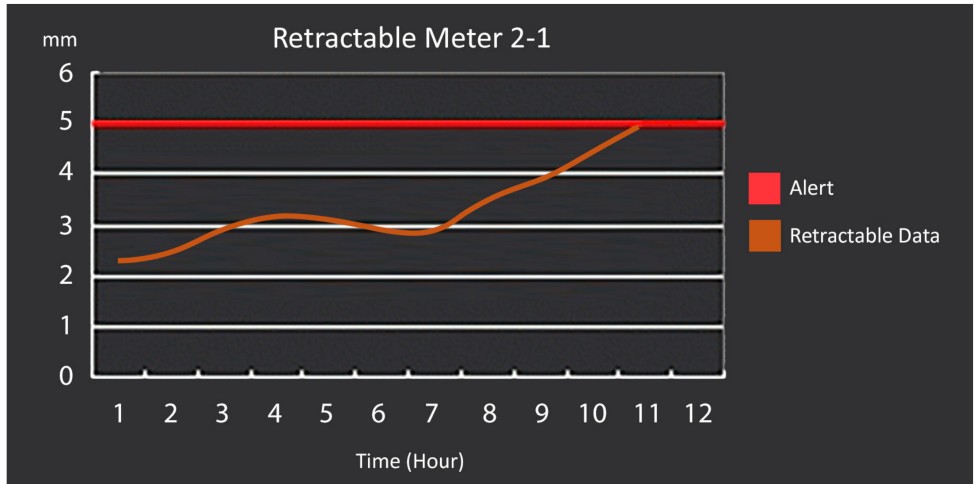

**Figure 7.** A detailed data display of the retractable meter.

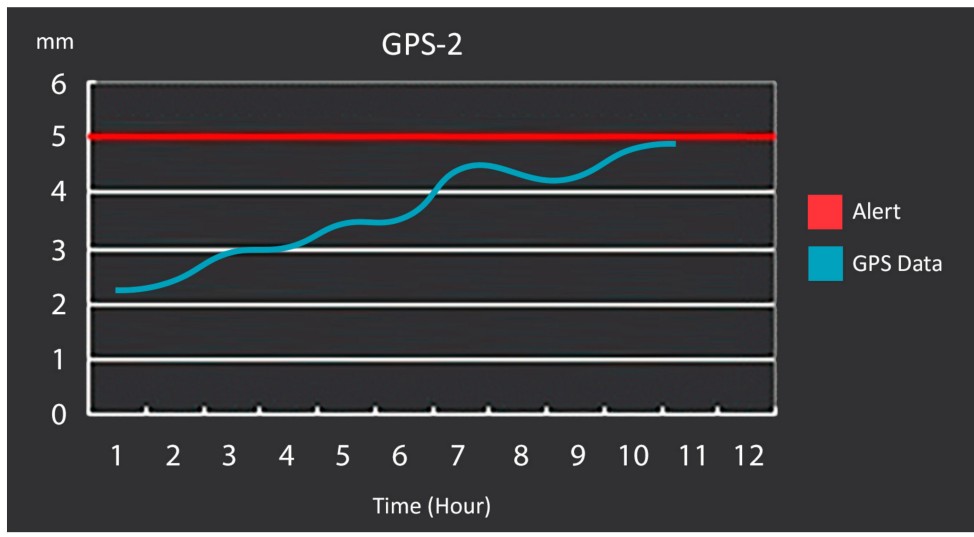

**Figure 8.** A detailed data display of the GPS.

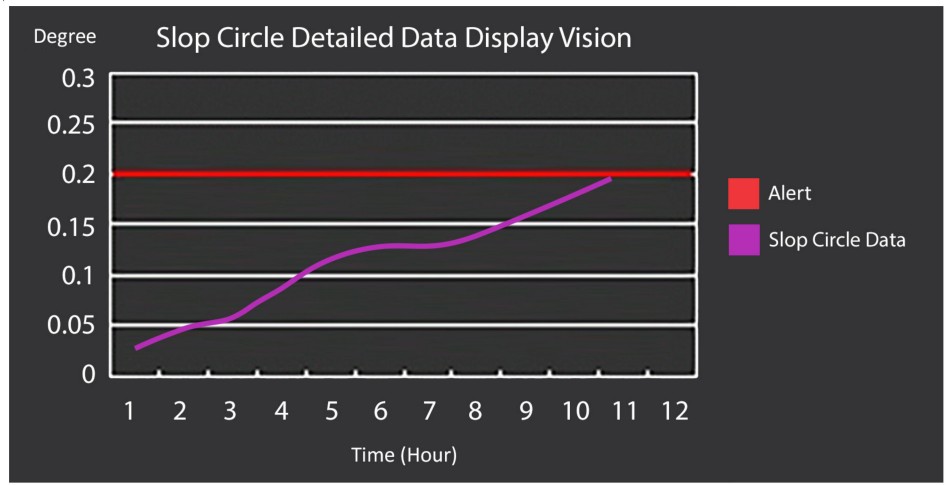

**Figure 9.** A detailed data display of a slop circle.

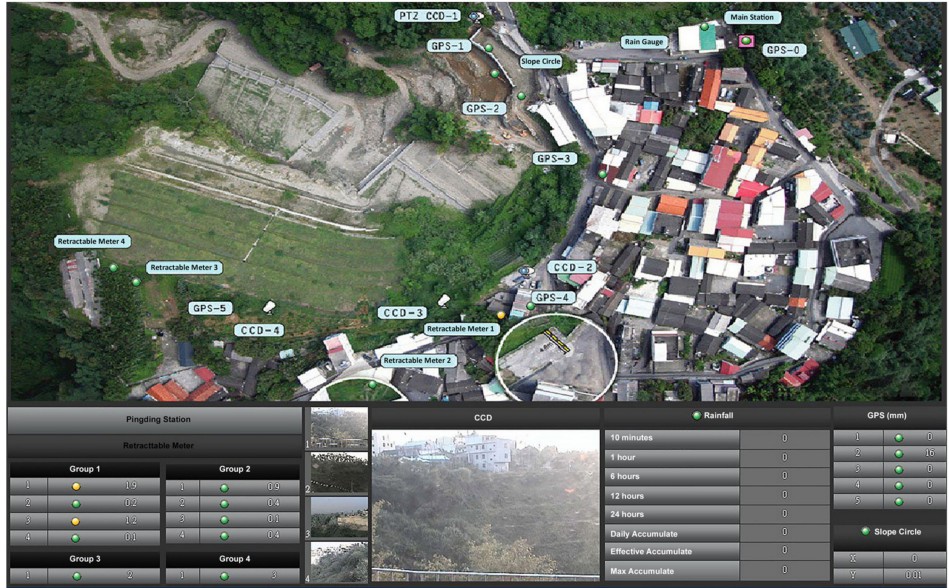

**Figure 10.** Integrated monitoring data display.

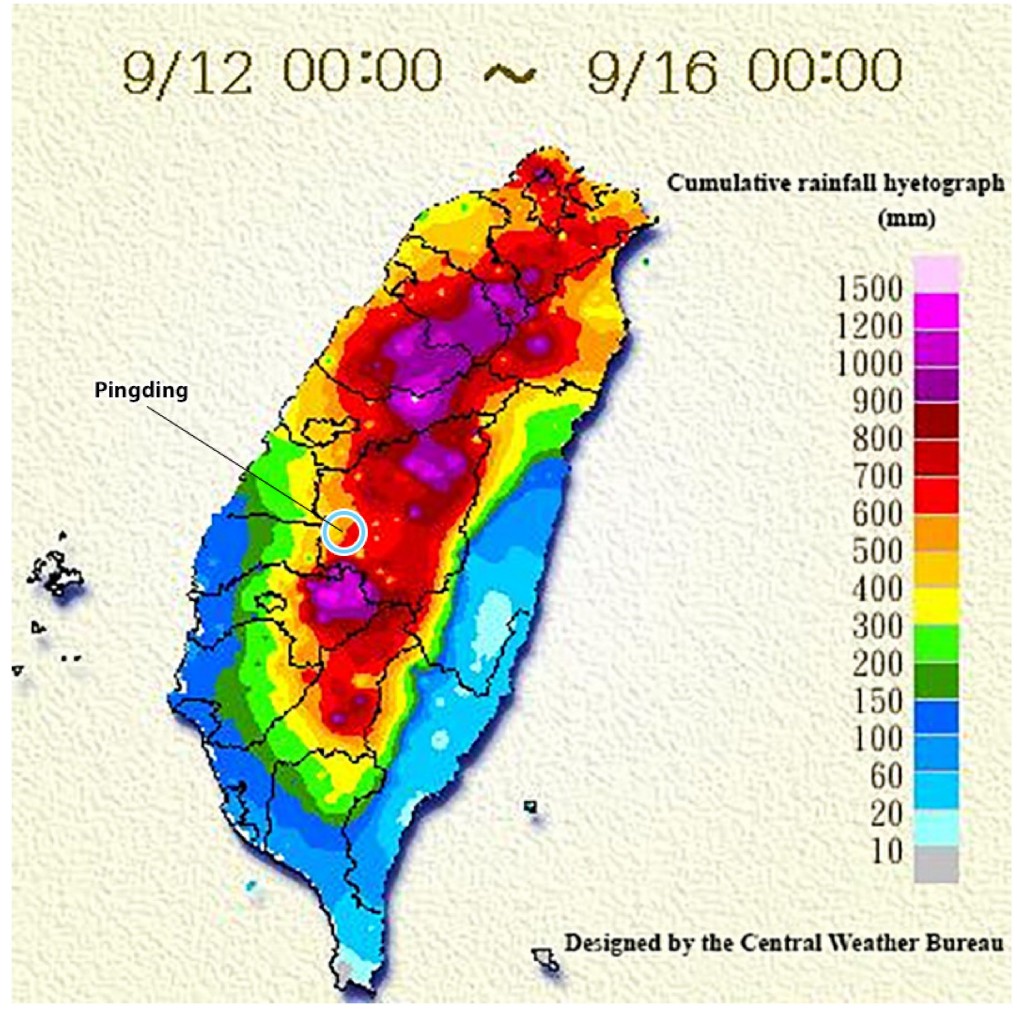

**Figure 11.** The cumulative rainfall from 12 to 16 September during Typhoon Sinlaku in 2008.

3.2.1. Rainfall Data

Influenced by Typhoon Sinlaku, the total cumulative rainfall from 11 September to 16 September at the observation stations was 349 mm, with the greatest rainfall intensity being 40.5 mm/h. A hyetograph of the cumulative rainfall during the time when the emergency response team was active, showing the rainfall intensity and the cumulative rainfall conditions, is provided in Figure 12. The yellow line in the figure represents the vigilance values, while the red line refers to the action values. The hourly rainfall reached 12 mm, and the daily cumulative rainfall was 86.5 mm at 6 a.m. on 14 September, which reached the threshold of the yellow level of vigilance. The hourly rainfall reached 20.5 mm, and the daily cumulative rainfall was 140.5 mm at 2 p.m. on 14 September, which reached the threshold of the red level of vigilance. No abnormal phenomena were observed in the comparison between the monitoring results of the GPS and retractable meters (Figure 13). The local village heads, the Yunlin County Government, and the related soil and water conservation departments were informed through the disaster report process of this system, as shown in Figure 14. It is assumed that the topographic features of the Pingding area, which increase the possibility of torrential rainfall within short periods of time, lead to higher rainfall intensity and heavier cumulative rainfall. Therefore, more data concerning local typhoons and torrential rainfalls will be collected in the future to determine whether the vigilance and action values of the rainfall in this area should be raised.

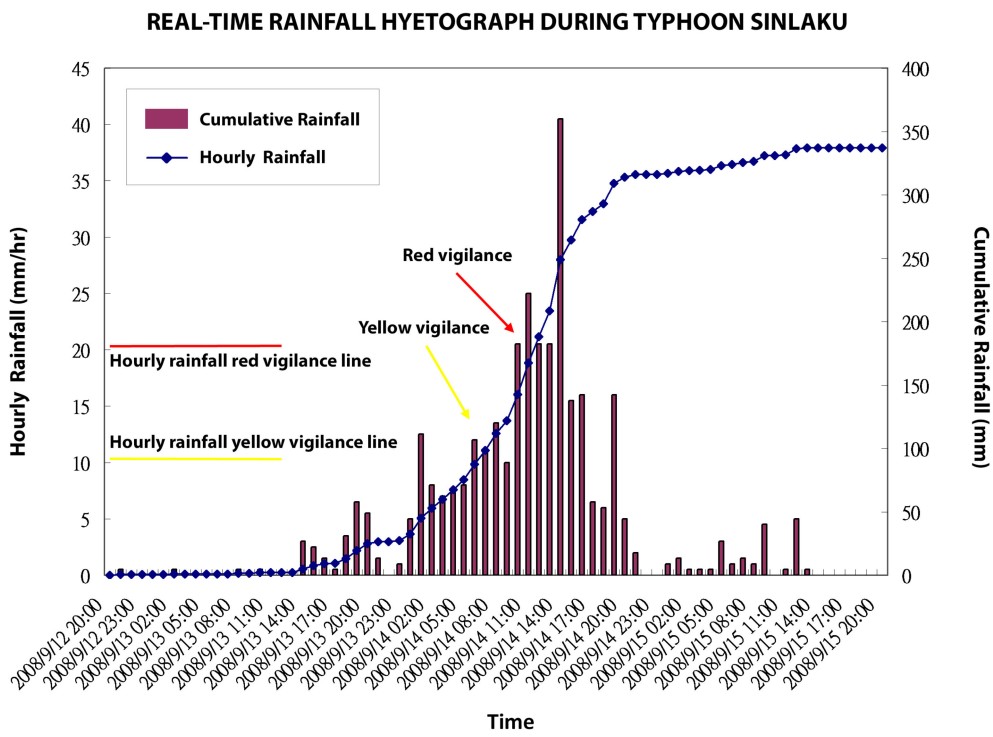

**Figure 12.** Real-time rainfall hyetograph during Typhoon Sinlaku.

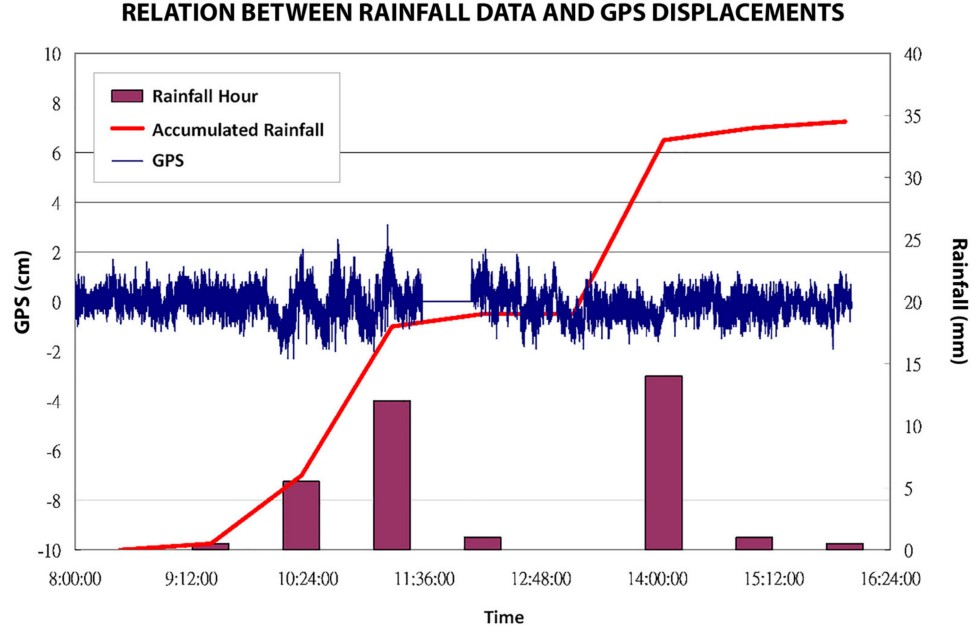

**Figure 13.** Relationship between Hourly Rainfall and GPS displacements.

### 3.2.2. GPS Monitoring Data

Figure 15 shows the GPS monitoring data in normal time without typhoons, as well as a comparison between the data and management values. As shown in the figure, all of the GPS points were in a stable condition, and the vibration of the displacement quantity was within 2 cm, indicating that its precision was approximately 2 cm, thus not exceeding the set threshold of the vigilance values. The GPS displacement of all of the action stations did not exceed the set threshold of vigilance values during Typhoon Sinlaku.

### 3.2.3. Monitoring Data of the Retractable Meters

The monitoring data from the magnetic induction and cable retractable meters during Typhoon Sinlaku, which are shown in Figure 16, were compared with the management values. Except for the micro-displacements caused by the indium steel wires of the cable retractable meters, all of the retractable meters remained in a stable condition and the set threshold of the vigilance values was not exceeded.

### 3.2.4. Slope Circle Monitoring Data

The slop circle monitoring data from the Typhoon Sinlaku period are displayed in Figure 17. The figure shows two bigger inclinations, which indicate soil retaining wall inclinations caused by the increase in soil pressure resulting from an increase in the water of the soil after rainfall. The data return to the initial position after the water in the soil evaporates or runs off. The data were compared with those of the management values, and all of the slopes were found to remain in a stable condition and the set threshold of the vigilance values was not exceeded. Figure 18 shows a detailed data display during Typhoon Megi—International Number 1013 is the 13th tropical cyclone named in the Pacific typhoon season in 2010, and was also the strongest storm of the year.

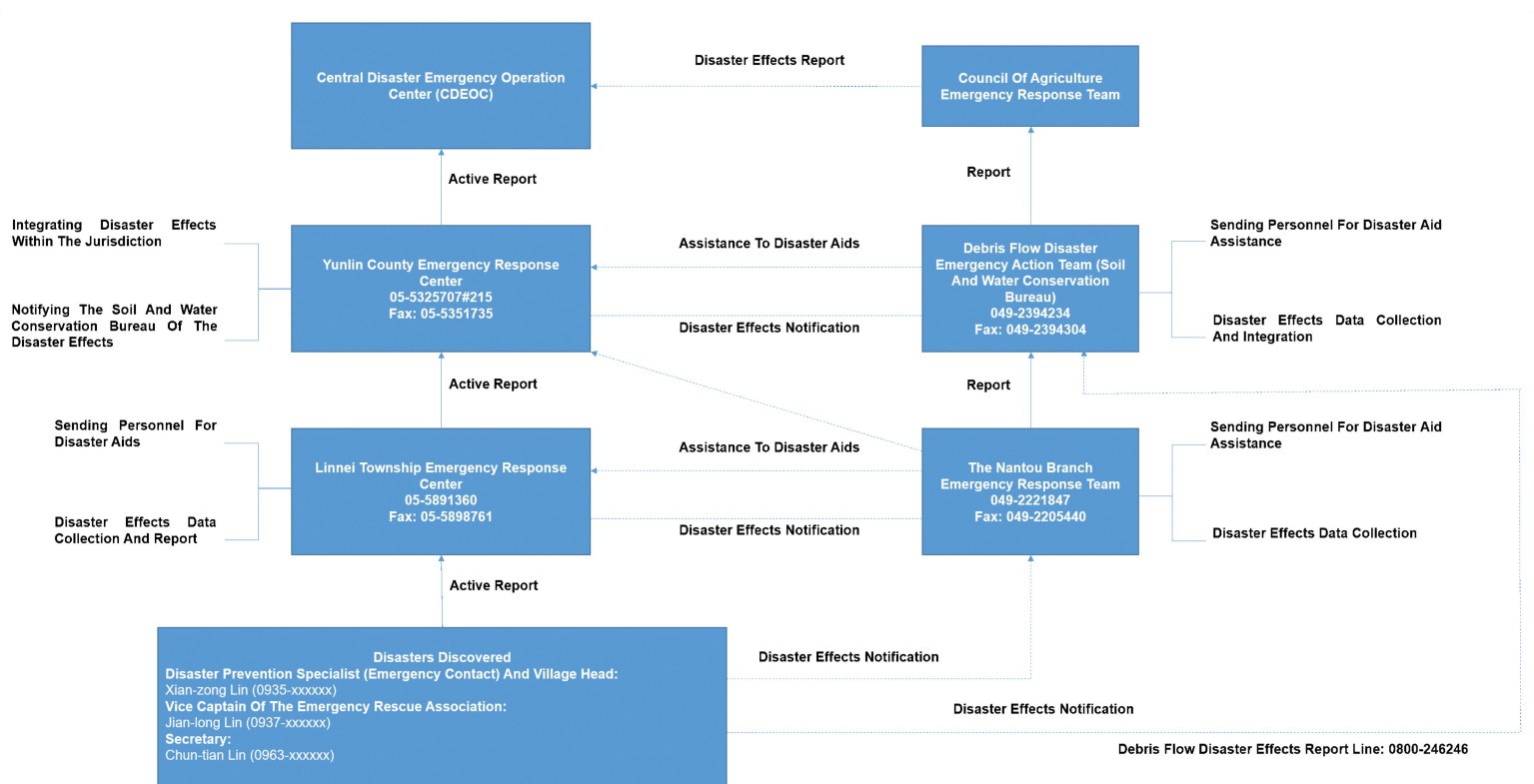

**Figure 14.** The reporting process for slope land collapse disasters.

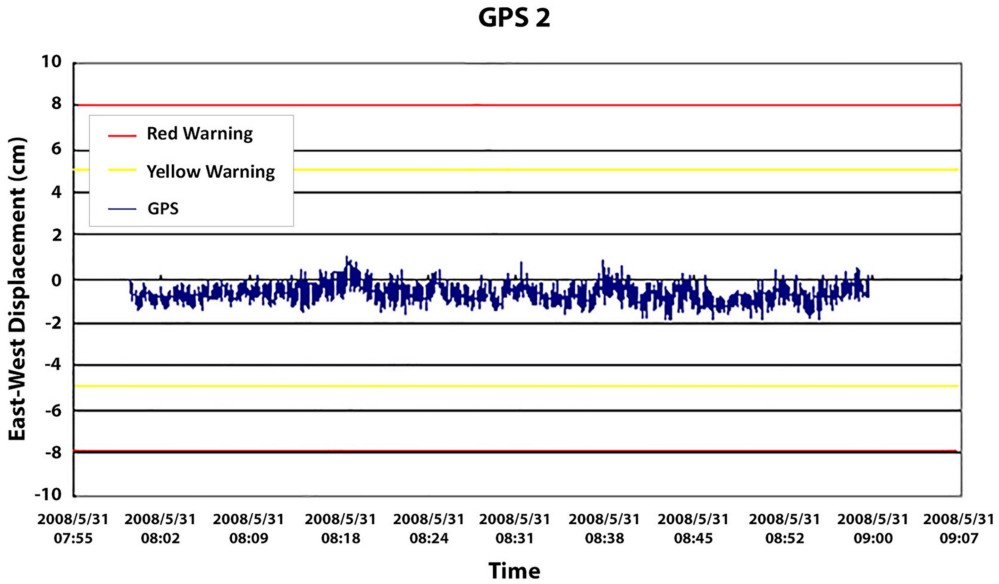

**Figure 15.** GPS-2 horizontal displacement quantities.

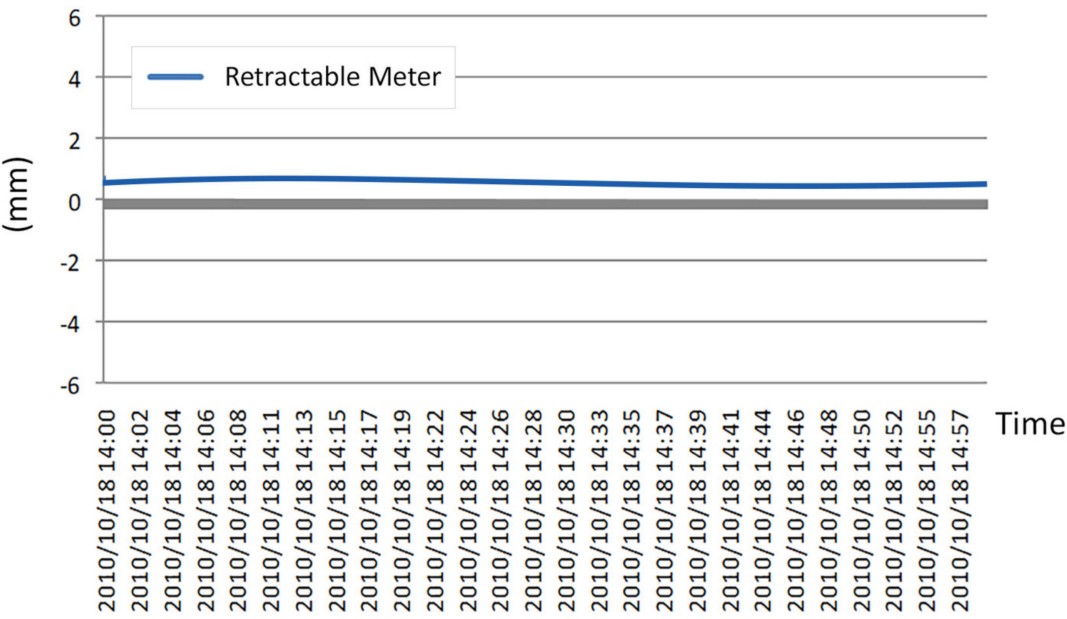

**Figure 16.** Detailed data display of the retractable meter during Typhoon Sinlaku (17–18 October 2010).

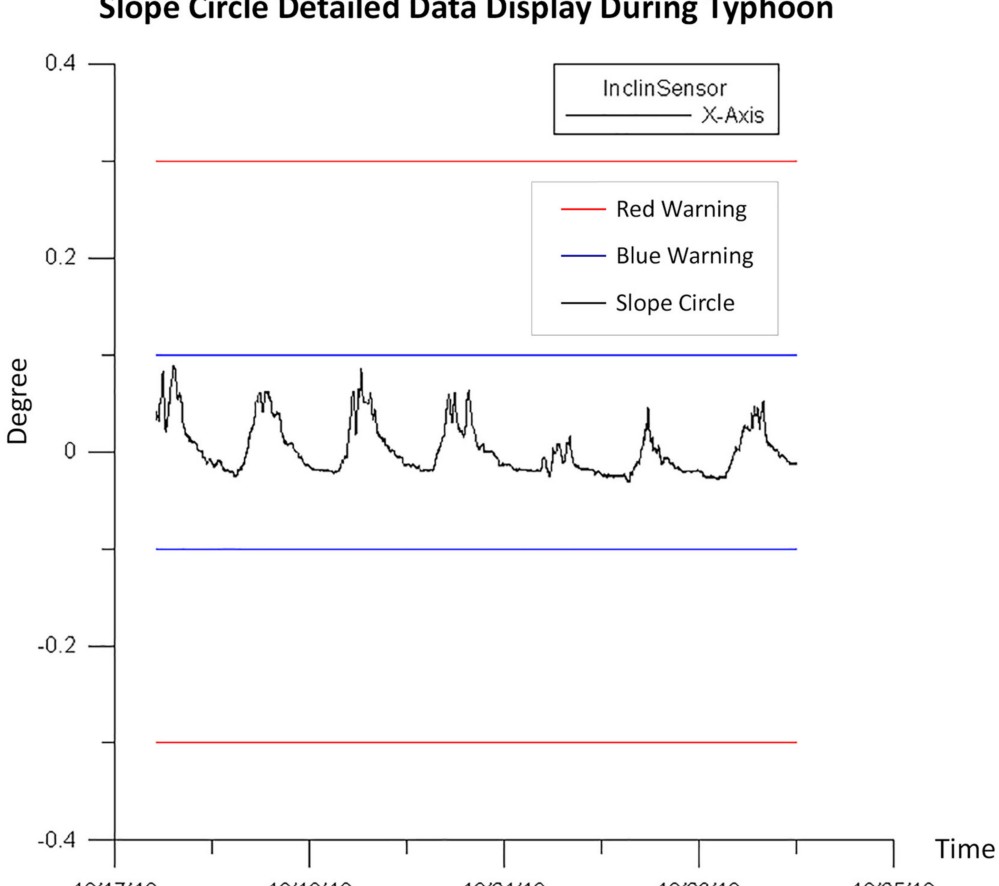

**Figure 17.** Detailed data display of the slop circle during Typhoon Megi (17–25 October 2010).

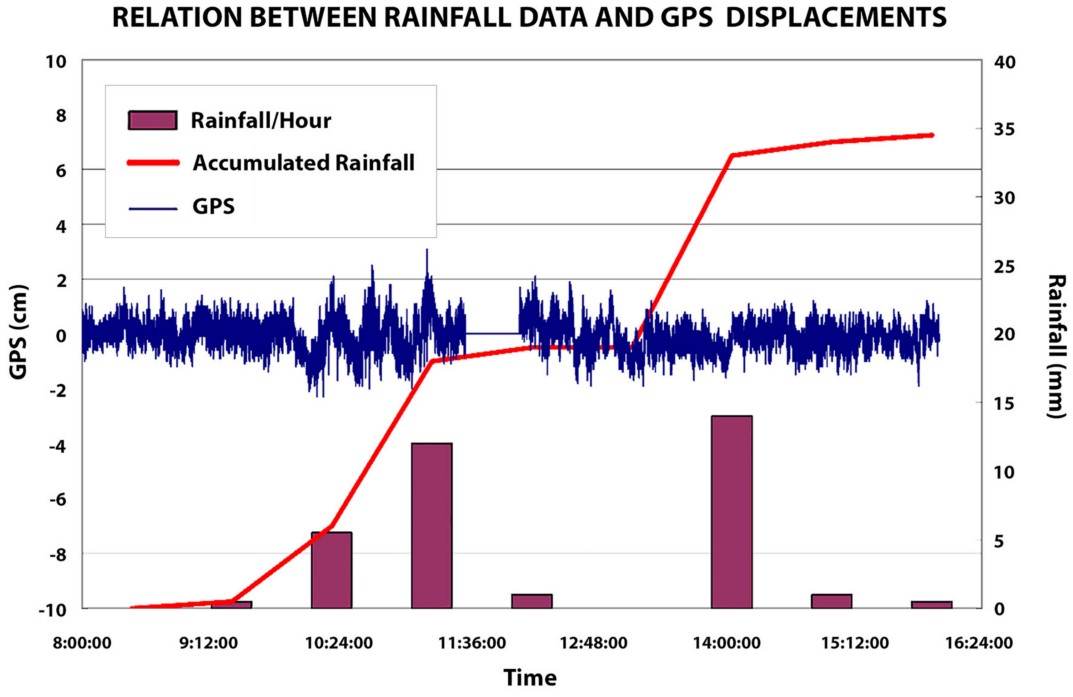

**Figure 18.** Relationship between rainfall and GPS displacement data.

### 3.2.5. Comparison between the Monitoring Devices and Rainfall

A synthetic comparison between the monitoring results is advised if the monitoring data surpass the threshold of the management values to determine whether a disaster has struck or whether an abnormal phenomenon of a single device has happened. As Figure 18 shows, a comparison between rainfall and GPS displacement involves determining the relationship between cumulative rainfall and displacement, as well as the relationship between rainfall intensity and displacement speed. Figure 19 shows a comparison between the CCD images of the time periods with the greatest rainfall intensity, aiming to ensure the rainfall monitoring is mistake-free.

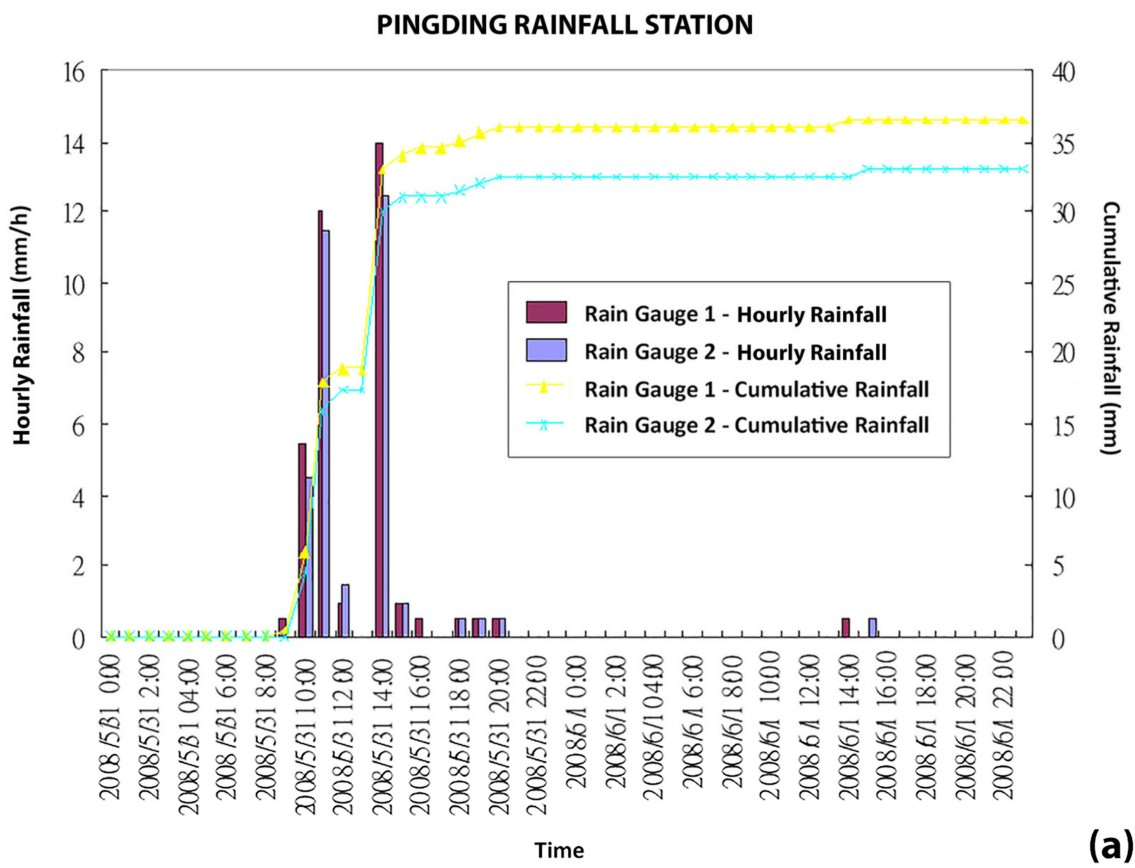

**Figure 19.** *Cont.*

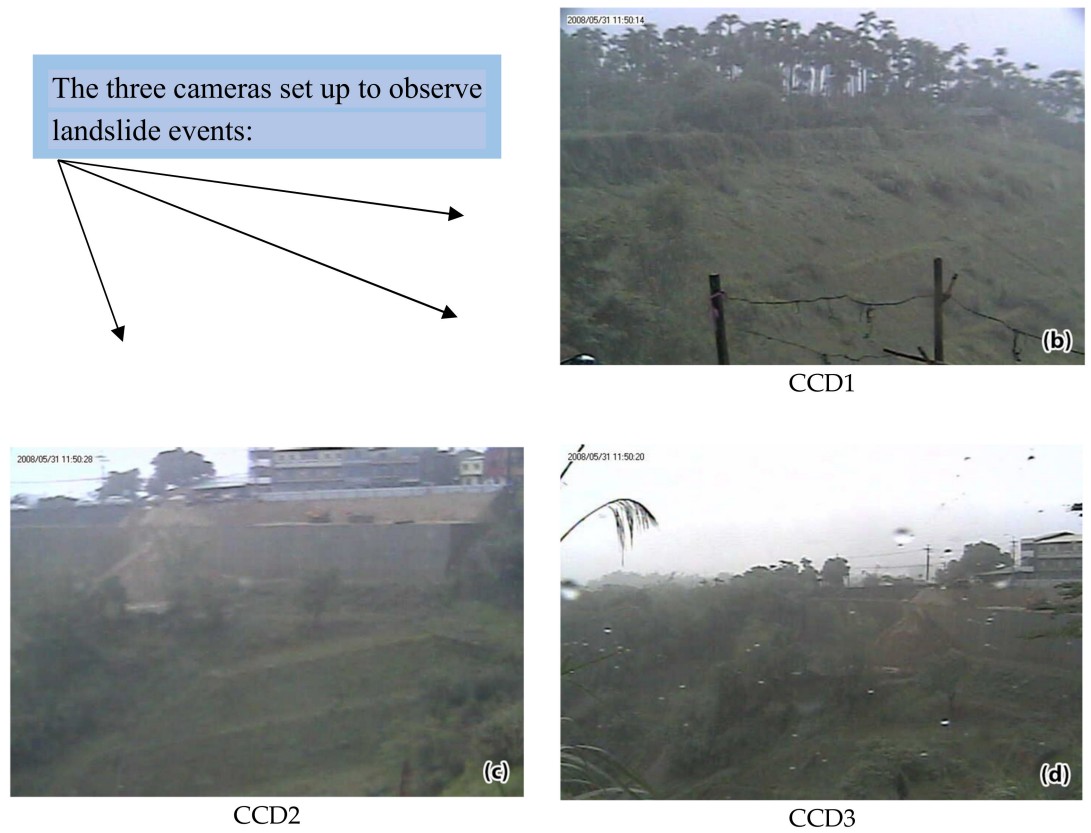

**Figure 19.** Relationship between rainfall and CCD data: (**a**) Cumulative rainfall of Pingding; (**b**) CCD camera 1; (**c**) CCD camera 2; (**d**) CCD camera 3.

During a typhoon and heavy rain, we can observe this system to determine the real-time accumulated rainfall intensity. This monitoring process demonstrated that there is no displacement of the ground or the slope of the retaining wall, which proves that our system is workable.

## 4. Discussion

According to the obtained rainfall data, Lumb established the scope of events of different gradations and provided an explanation based on 15-day pre-storm rainfall and 24 h rainstorms. Severe events brought over 100 mm of rainfall in 24 h and over 350 mm of pre-storm rainfall in 15 days, while serious events generated more than 100 mm of rainfall in 24 h and 200 mm of pre-storm rainfall. This proof of a relationship between side slope failures and rainfall encouraged many studies focusing on the threshold of rainfall for causing side slope failures. For example, it was determined that the mechanism of side slope failures is successfully triggered when the rainfall in Southern California reaches 140% of its normal rainfall (an average value of the recorded data of over 100 years) [34–36], suggesting that the threshold for Los Angeles is 125% (1993), which was also confirmed by Wieczorek [37]. Auer and Shakoor studied the collapse distribution of Nelson County, central U.S., after Hurricane Camille in 1969. According to their research, the hurricane, which moved from the west to the east, caused a more serious collapse of the side slopes in the west, northwest, and southwest directions. This indicates that there might be a connection between the directions of side slopes in collapse zones and hurricane routes [38]. Hong Kong has no records of collapse generated by earthquakes; however, the Geotechnical Engineering Office compared the risk of earthquakes and rainfall in terms of their generation of artificial slope collapse [39]. The study results showed that earthquakes bring a lower risk of causing artificial slope collapse than rainstorms.

Brand, Permchitt, and Phillipson [40] examined the side slope failures of Hong Kong through the following three steps:

- The data provided by Lumb [32] were referred to and the rainfall data collected through Hong Kong's 46 automatic rainfall recorders were recorded;
- The considered rain delay was 1 h and 24 h of rainstorms, which was matched with the 30-day pre-storm rainfall;
- The timing of side slope failures was documented based on the reports of the National Fire Agency.

According to the above research, based on the relationship between disaster numbers and rain intensity, the authors [40] concluded that over half of the studied side slope failures in Hong Kong were caused by local brief rain showers, and that the threshold for triggering side slope failures is crossed when the rain intensity reaches 70 mm/h.

Shi-Jie Jian [41], who discussed the influences of different factors on the mechanism of side slope slips through on-site measurement data, focused on the slopes (28 K + 900 to 31 K + 500) along the roads near Wuwanzi in Gongtian village, which is in the township of Fanlu in Jiayi county. As part of the old landslip region, this area has suffered from landslips for a long time—since the opening of the road. Several measurement devices have been installed for two-year periods since the beginning of 2000, carrying out consistent observations of local strata, land surface deformation, the underground water level, and rainfall. Relevant discussions have also been carried out in coordination with basic theory and indoor experiments. According to the observation results, the condition of this particular area is extremely unstable. It has also been discovered that the slip behaviors of this area are highly related to rainfall. The monitoring data since 2000 indicate that the cumulative rainfall needed for accelerated side slope slips is 80–270 mm, which equals a cumulative rainfall of 3–5 days.

Xing-Fu Ye [42] discussed the influence of rainfall seepage recharge on slope land collapse through the concept of watershed-based water balance using the following steps: (1) Estimating the base flow volume through the river flow volume qualification line method and the base flow estimation mode, and related to base flow as the seepage recharge of underground water; (2) taking into consideration the correlation system of rainfall, seepage, run-off, evapotranspiration, and underground water recharge through the unsaturated soil water balance method. These two modes were both shown to have similar results. The next step was to carry out a sensitivity analysis through STEDWIN (STEDwin is the smart editor to simplify working with Purdue or PennDOT* STABL programs) and to discuss its influence on the side slope stability. This revealed that the internal friction angle variability affects the side slope stability most significantly, followed by sloped variability and the rise of the underground water level, while the coherence and unit weight come next. From the figure of the area's rainfall and safety factor relationship, it can be seen that as the steepness of the slope reaches 35 degrees and the rainfall reaches 400–500 mm, according to the analysis of Janbu's simplified method, the safety factor reaches 1.0, forming an imminent risk of collapse. Regarding the discussion of practical examples, a correlation was found between the locations in the Cingshuei River basin that have suffered from collapse and the estimated rainfall recharge factors.

Yi-Feng Qiu [43] discussed the effects of the relevant factors that have affected the mechanism of side slope slips through on-site measurements and theoretical analyses. Long-term monitoring work regarding information such as land surface and stratum heterotaxies, rainfall, the underground water level, and the underground water flow volume of the surrounding areas of Wuwanzi along the Taiwan 18th Line Highway (29 K + 900 ~ 31 K + 500) has been carried out since the beginning of 2000. Serious failures took place on 26 June 2003, leading to the depletion of 150 m of the road base of the 31 K + 340 section and a 1.5-month highway disruption. The causes of this event's side slope failures were analyzed, discussed, and compared with the on-site monitoring results. With the theoretical analyses and the estimation of the stability of falling residual slopes, a final possible restoration solution was proposed. Ming-ren Gao [44] studied the slope lands near Wuwanzi along the Taiwan 18th Line Highway. An unexpected slip failure struck the 31 K + 340 section on 26 June 2003, causing disruption of the entire highway. A simulation of dry seasons and rainy seasons based on the underground water level was conducted using SLOPES/W software (SLOPE/W is the leading slope stability software for soil and rock slopes, GeoStudio company, head office in Calgary, Alberta, Canada). The load input was

increased during rainy seasons to achieve the side slope's failure conditions. The failure mode was then integrated, analyzed, and compared with the observation statistics of the tiltmeter.

The Japanese Society for Landslip Solution Technologies [45] suggested a classification of the displacement speed effect on unstable side slope activities, as shown in Table 2. Such a classification is the displacement speed obtained through dividing the obtained stratum displacement by the observation time. The basis of classification depends on the level of speed and the trend of displacement. The table shows that unstable side slope activities can be roughly classified into four categories. The first three categories are side slopes with (1) emergency movements, (2) definite movements, and (3) semi-definite movements, which are moving slopes with confirmed problems that require immediate effective solutions. The fourth type of movement suffers from potential hazards in terms of the stability of side slopes if there is a certain trend in the displacement direction, because the displacement speed of this type of potentially moving side slope is extremely low—i.e., 0.5–2.0 mm every month, thus producing a displacement quantity of 6–24 mm every year. Further measurements are needed to confirm the stability of side slopes with displacement speeds of over 0.5 mm/month but without certain cumulative displacement directions (forward displacements and backward displacements have both been observed). The stability of such side slopes can be affected by measurement errors of devices or by the non-compact surrounding stuffing of inclination observation tubes when installed. The relevant precaution solutions and management principles provided by the Japanese Society for Expressways [46] are displayed in Table 3.

**Table 2.** Table of displacement speed and side slope stability judgment and suggestions [45].

| Movement Types | Daily Movement (mm) | Monthly Movement (mm) | Accumulative Inclination of Certain Directions | Activity Judgement | Abstract |
|---|---|---|---|---|---|
| Emergency movements | Over 20 mm | Over 500 mm | Significantly apparent | Rapid mass wasting | Mass wasting movements and mudslides |
| Definite movements (movement values) | Over 1 | Over 10 | Apparent | Moving actively | Colluvial soil slips and deep slips |
| Semi-definite movements (vigilance values) | Over 0.1 | Over 2.0 | Slightly apparent | Moving slowly | Clay slips and backfill slips |
| Potential movements (alarm values) | Over 0.02 | Over 0.5 | Small | Further observation is needed | Clay slips and deep slips |

**Table 3.** Monitoring management benchmarks of stratum slips [46].

| Monitoring Methods | Management Classification | | |
|---|---|---|---|
| | Attention | Vigilance | Evacuation Required |
| Land surface retractable meters | | >10 mm/day | >50 mm/day |
| Rainfall intensity | 0.5–10 mm/day | 10–20 mm/h | >20 mm/h |
| Cumulative rainfall | | 50 mm | 100 mm |

To date, no domestic regulations have been set for relevant management values; in addition, no long-term monitoring results have been gained. Consequently, referring to Japan's recommended values and the device-measurable precision of side slope stability, judgement is inevitable, since Japan already has a certain amount of related experience. The retractable meters' action values are 5–10 times that of the vigilance values. The tentative monitoring management values are shown in Table 4 based on the premise that all of the monitoring devices' action values are five times that of their vigilance values. In the future, the data obtained by the monitoring of the devices will be analyzed, interpreted, and judged, and interactive comparisons will be made between different monitoring devices.



**Table 4.** Management values of the monitoring devices [30].

| Monitoring Devices | Rain Gauge, Rainfall Intensity, and Cumulative Rainfall | Retractable Meters (Magnetic Induction) | Retractable Meters (Cable) | GPS (Horizontal) | Slope Circles |
|---|---|---|---|---|---|
| Action values (red light) | >20 mm/h and 100 mm/day | >±50 mm/day or ±5 mm/min | >±250 mm/day or ±25 mm/min | ±80 mm/min | ±0.3°/min |
| Vigilance values (yellow light) | 10–20 mm/h and 50 mm/day | >±10 mm/day or ±1 mm/min | >±50 mm/day or ±5 mm/min | ±50 mm/min | ±0.1°/min |
| Precision | 0.5 mm | 0.1 mm | 1 mm | 30 mm | 0.01° |

In this study, by using these sensors, we integrated software and hardware to monitor landslides in Pingding. We aimed to review the benchmarking management values and to further investigate the management values. In addition, improvement of the real-time monitoring system was also expected after amending the management values based on different viewpoints to better determine the effects and efficiency of monitoring and vigilance.

In Pingding, Taiwan, particularly in summer (i.e., the rainy season), rainfall is the main contributing factor to the occurrence of landslides, rockfalls, and mudflows. Rainfall information is mainly used to monitor whether there is rain alert on the spot. Station rain gauges, automatically adopt the rain station's operation method, and provide a self-sufficient method. Such stations play a role in recording data, storage, transmission, and display systems for monitoring landslides. The results of this research not only monitor the landslide in real-time, but also manage the space transformation conditions when disasters occur.

The advantages of such a station are: (1) The station is equipped with advanced sensors and technologies; (2) provision of early warning information in real-time for residents who live in the area, as well as for managers who are in charge of this issue, by sending an SMS message to a mobile device; (3) a successful landslide monitoring station can support the monitoring network in Taiwan.

The feasibility of monitoring collapses and displacements through GPS was proven by this study. In addition, with the precision of different monitoring devices, the judgment of the management benchmark values of collapse monitoring established by domestic and overseas documentations was found to be reasonable, according to the monitoring results. Further data concerning typhoons and torrential rainfall will be collected in the future to confirm or amend the management benchmark values.

Finally, this study plays a vital role in the early warning system of landslides in Pingding, as highly effective monitoring can provide information on the current status that is not only applicable for weather forecasts, but also for disaster mitigation policy.

## 5. Conclusions

This study established the Pingding Monitoring System in the surrounding area of the Pingding Collapse Zone. On-site observation equipment was installed, and topographic and hydrological information was also collected as a reference for local disaster prevention and response.

Centered on emergency response, this study mainly focused on the displacement measurement of land surfaces. In light of the analysis results of side slope stability, the main cause of collapse and slips appears to be the water content of the soil. The previous collapse disaster confirmation suggests that inbuilt meters that can detect the soil moisture in mudstones should be used to serve as a reference for the necessity of advanced vigilance or as an alert when the water content in the soil is on the rise.

Regarding modern disaster prevention management thinking, the strategy for tackling disasters is a process that involves the application of the following four steps: (1) Readiness—establishing emergency response measures and a management system to enable a quick response once a disaster strikes; (2) response—taking immediate actions before, during, and after disasters to reduce the number of casualties and the property loss and to accelerate recovery; (3) restoration, including the restoration of basic vital resource supply systems within short periods of time and the long-term responsibility of restoring people's normal lives; (4) disaster reduction—advocating for certain policies and applying

particular measures to alleviate the impacts brought by future disasters. The above thinking suggests that solving disaster management problems through single modes is not sufficient; instead, in addition to traditional engineering treatments, it is essential to consider multilateral aspects such as multilateral and cross-field discussions involving the following issues: Climatology, economics, engineering, geography, geology, law, meteorology, planning, psychology, public policy, and sociology.

In the future, we would like to apply this research to further applications of the system and its subsequent monitoring in order to establish an early-warning system (social-economic goal) and to conduct post-event analysis when a landslide has actually occurred (scientific goal).

**Author Contributions:** Conceptualization, T.Y.C. and T.V.H.; data curation, Y.M.F.; formal analysis, Y.M.F. and T.V.H.; funding acquisition, T.Y.C. and Y.M.F.; investigation, Y.M.F. and Q.T.B.; methodology, Y.M.F. and Q.H.N.; project administration, T.Y.C. and Y.M.F.; software, Y.M.F. and Q.H.N.; supervision, T.Y.C. and D.B.N.; validation, T.Y.C. and D.B.N.; visualization, Y.M.F. and T.V.H.; writing—Original draft, T.-Y.C. and T.V.H.; writing—Review and editing, T.Y.C. and T.V.H. All authors have read and agreed to the published version of the manuscript.

**Funding:** This article is the result of a state-level project financed by the Soil and Water Conservation Bureau and Geographic Information Systems Research Center, Feng Chia University, Taiwan (grant number MOST20051110).

**Acknowledgments:** The authors thanks the Soil and Water Conservation Bureau and Geographic Information Systems research Center, Feng Chia University, Taiwan for their support.

**Conflicts of Interest:** The authors declare no conflict of interest.

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
