# Peer review of "New Landslide Disaster Monitoring System: Case Study of Pingding Village"

_applsci, doi:10.3390/app10196718_

Round 1

Reviewer 1 Report

This article is well written and the methodology employed is quite original. It is serious work which however deserves some adjustments because the "discussion" part is not sufficiently developed. I therefore recommend to the authors to densify their analysis at the level of the "discussion" part to allow the rapid publication of this quite interesting work.

Author Response

This article is well written and the methodology employed is quite original. It is serious work which however deserves some adjustments because the "discussion" part is not sufficiently developed. I therefore recommend to the authors to densify their analysis at the level of the "discussion" part to allow the rapid publication of this quite interesting work.

Response: Dear Reviewer: We have wrote more for the chapter Discussion, with red color.

We are sincerely thank you so much for your time and your positive comments.

Wish you all the best!

Best regards,

Authors team.

Reviewer 2 Report

This manuscript will be interested to the readers Applied Sciences. However, in my opinion this document needs to be improved before publication. In fact, some parts on the text read like a technical report rather than an innovative scientific contribution.

General Comments

  1. The introduction section needs to be revisited and made more concise. The authors need a background section where they dig a bit deeper into the high-level topics that were introduced in the introduction (overview of the current status). I suggest you include more references and the authors’ contribution to the state of the art.
  2. A flowchart describing the whole procedure proposed by the authors will be useful for the readers to have a quick overview of this work.
  3. For figures with more than one panel, panels should be clearly indicated using labels (a), (b), (c), (d), etc. However, do not embed the panel labels over any part of the image but place them in a vertex of figure. Avoid using "bottom", "top" etc. to panel labels. Any individually labeled figure parts or panels (a, b, etc.) should be specifically described by part name within the caption.
  4. The figures in general are not very scientific, I strongly advise you to redo the figures from 5 to 8.
  5. The paper is not well-structured: The materials and case study, the methodologies applied, its results and conclusions should be clearly distinguishable. Line 294 “Example…” You cannot give an example, you must describe the method and then the results of the method applied to the case study.

Specific comments:

L.33 - L37: This part is very poor, many methodologies have been studied.

L.74: please specify the reference system you are using.

L.104: Figure 1, in line 74 you talk about geographical coordinates, then in figure 1 you use projected coordinates (UTM?), please eliminate zeroes in the decimals of the coordinates and specify the reference system in the caption. See point 3 of the general comments.

L.150: I suggest you to first make a summary of the methods used, also for points, inserted here a flowchart describing the whole procedure proposed.

L.173: the figure 3 is not very readable, improves the resolution.

Author Response

General Comments

  1. The introduction section needs to be revisited and made more concise. The authors need a background section where they dig a bit deeper into the high-level topics that were introduced in the introduction (overview of the current status). I suggest you include more references and the authors’ contribution to the state of the art.

Response 1: Dear reviewer 2: we have wrote more more references follow your suggestions.

For the authors’ contribution, we have indicate our contribution in the Line 72 we are Feng Chia University. We carried out this project.

  1. A flowchart describing the whole procedure proposed by the authors will be useful for the readers to have a quick overview of this work.

Response 2: Dear Reviewer: as your suggestion, we draw a flowchart to describe the whole process (Figure 1 )

  1. For figures with more than one panel, panels should be clearly indicated using labels (a), (b), (c), (d), etc. However, do not embed the panel labels over any part of the image but place them in a vertex of figure. Avoid using "bottom", "top" etc. to panel labels. Any individually labeled figure parts or panels (a, b, etc.) should be specifically described by part name within the caption.

Response 3: Dear Reviewer: Yes, we have re-draw to indicate using labels (a), (b), (c), (d) in the figure 2, and Figure 20.

  1. The figures in general are not very scientific, I strongly advise you to redo the figures from 5 to 8.

Response 4: Dear Reviewer: We have re-draw new figure from 5 to 8 follow your suggestions.

  1. The paper is not well-structured: The materials and case study, the methodologies applied, its results and conclusions should be clearly distinguishable. Line 294 “Example…” You cannot give an example, you must describe the method and then the results of the method applied to the case study.

Response 5: Dear Reviewer: Yes, we have edited and change the title to “Results”

Specific comments:

  • 33 - L37:This part is very poor, many methodologies have been studied.

Response 1: Dear Reviewer, we have wrote more and cite more as your suggestions. Thanks so much.

  • 74: please specify the reference system you are using.

Response 2: Dear Reviewer: yes, we follow the MDPI format by the ACS style, is there any mistakes, please feel free to let me know. Sincerely thank you so much!

  • 104: Figure 1, in line 74 you talk about geographical coordinates, then in figure 1 you use projected coordinates (UTM?), please eliminate zeroes in the decimals of the coordinates and specify the reference system in the caption. See point 3 of the general comments.

Response 3: Yes, that one we use Projected Coordinate System: TWD_1997_TM_Taiwan

  • 150:I suggest you to first make a summary of the methods used, also for points, inserted here a flowchart describing the whole procedure proposed.

Response 4: Thanks so much for your suggestion, we have new figure a flowchart to describe the whole procedure (Figure 1)

  • 173: the figure 3 is not very readable, improves the resolution.

Response 5: Dear Reviewer, yes, we has changed new one, we did capture it from the system user interface. Hopefully you can accept that for us!

We are sincerely thank you so much for your time and your positive comments.

Wish you all the best!

Best regards,

Authors team.

Reviewer 3 Report

Ref:       applsci-920913

Title:      New Landslide Disaster Monitoring System: Case Study of Pingding Village

Journal: Applied Science

As I am not native speaker I think that I am not the right person for judge of the language level or quality.

The paper deals with an interesting theme in the context of prevention of specific type of disasters which can cause human lost of human lives. It is really important to search best applicable prevention solution which can minimise this risk. The title of the paper is acceptable and adequate and no changes are necessary. I find the abstract relatively acceptable and structured, but in my view more information about results and recommendation should be added. In keywords should be used another words than are used in the title, in case that you are using the same words then decreasing possibility that your article going to be find, because for searching I can use title or keywords.

The manuscript has a sufficient scientific value and the information provided represents widening of knowledge. However, in my view the structure of the manuscript have to be rebuilt, especially chapter Material and methods and Discussion. Discussion seems to me as a big weakness of this article, there are no critical comparison of results or factors which affect the results with results or recommendations another authors. There is no citations in your chapter Discussion, so it is not a discussion but only your statement without any comparison.

Lines 70 – 71 are without citation so it seems that text belong to your work. In the end of Introduction the aim of your study should be place.

Chapter Material and Methods should be structure for example like as follow:

  1. Material and Methods

2.1 Study area

2.2 Data collection

2.3. Data source and their processing

2.4 Data analysis

All images like maps or aerial photos should be accompanied by a scale (numeric or graphic)

Lines 194 – 290 seems to me like text which should be used in chapter Discussion not in Material and Methods

.

Author Response

The paper deals with an interesting theme in the context of prevention of specific type of disasters which can cause human lost of human lives. It is really important to search best applicable prevention solution which can minimise this risk. The title of the paper is acceptable and adequate and no changes are necessary. I find the abstract relatively acceptable and structured, but in my view more information about results and recommendation should be added. In keywords should be used another words than are used in the title, in case that you are using the same words then decreasing possibility that your article going to be find, because for searching I can use title or keywords.

The manuscript has a sufficient scientific value and the information provided represents widening of knowledge.

Point 1: However, in my view the structure of the manuscript have to be rebuilt, especially chapter Material and methods and Discussion. Discussion seems to me as a big weakness of this article, there are no critical comparison of results or factors which affect the results with results or recommendations another authors. There is no citations in your chapter Discussion, so it is not a discussion but only your statement without any comparison.

Response Point 1: Dear Reviewer: Thank you so much for your suggestions, we have rewrite and re-arrange for the part of Discussion, with more citation (Line 336- Line 462)

Point 2: Lines 70 – 71 are without citation so it seems that text belong to your work. In the end of Introduction the aim of your study should be place.

 Response 2: Dear Reviewer, we have rewrite and add the aim of this study. Thank you J

Point 3: Chapter Material and Methods should be structure for example like as follow:

  1. Material and Methods

2.1 Study area

2.2 Data collection

2.3. Data source and their processing

2.4 Data analysis

 Response 3: Dear Reviewer, we have re-arranged this part follow your suggestions.

Point 4: All images like maps or aerial photos should be accompanied by a scale (numeric or graphic)

Response 4: Dear Reviewer, we add graphic for those maps

Point 5: Lines 194 – 290 seems to me like text which should be used in chapter Discussion not in Material and Methods.

Response 4: Dear Reviewer, we have moved Lines 194-290 to the chapter Discussion.

We are sincerely thank you so much for your time and your positive comments.

Wish you all the best!

Best regards,

Authors team.

Round 2

Reviewer 2 Report

The authors have done a good job addressing all comments and I have no further big suggestions. I believe the paper is acceptable for publication in the Applied Sciences Journal.
Anyway, I find the figures still not very "scientific", for example figure 8, 9, 10 and 12.
Flowchart are too simple; I suggest you improve them.
Line 98, figure 1? I still read different coordinates in the figure!

Author Response

Response to Reviewer 2 (Round 2)

Comments and Suggestions for Authors

The authors have done a good job addressing all comments and I have no further big suggestions. I believe the paper is acceptable for publication in the Applied Sciences Journal.

Point 1: Anyway, I find the figures still not very "scientific", for example figure 8, 9, 10 and 12.

Response Point 1: Dear Reviewer: those figures we would like to prove the station is working: During the landslide event, all the sensor's data have the same trend. The retractable meter, GPS, and Slip circle reach the alert at the same time, as shown in Figure 8-10.

Figure 12: based on the analysis data, we would like to show the rainfall intensity and the cumulative rainfall conditions during that typhoon even.

Point 2: Flowchart are too simple; I suggest you improve them.

Response Point 2: Dear Reviewer, we have re-drawn the new one as Line 183 (Figure 4)

Point 3: Line 98- figure 1? I still read different coordinates in the figure!

Response Point 3: Dear Reviewer, we have re-drawn the new one (Line 127- Figure 2)

Dear Reviewer,

We are sincerely thank you so much for your support and your positive comments. Please accept my deepest thanks.

Wish you all the best!

Best regards,

Author team.

Reviewer 3 Report

In this time I do not have any others comments.

Author Response

Reviewer 3 (Round 2)

Comments and Suggestions for Authors

In this time I do not have any others comments.

Dear Reviewer,

We are sincerely thank you so much for your support and your positive comments. Please accept my deepest thanks.

Wish you all the best!

Best regards,

Author team.

This manuscript is a resubmission of an earlier submission. The following is a list of the peer review reports and author responses from that submission.

Round 1

Reviewer 1 Report

The submitted version of the paper is far from the standard of a scientific paper, thus it must be near completely re-written.

Several issues of the introduction (geological setting, the "921 earthquake, and so on) lack in References. Where did the authors take the term "heterotaxy", in relation to the landslide? The matter is obscure to me. Moreover I suggest to move many questions in the "study area", writing the "introduction" according to the "instruction for authors" of Water.

The figures are, almost all, of a very scarce quality. Figure 1 is even unreadable due to an overlap of the legend. In the rewriting of the paper, the quality and significance of the images must be treated with great care. Figures 5-7 must be improved in readability. The pictures of the instruments (Figures 9-14) are useless. There are two figures numbered as "29". Again, Figure 14 portraits an inclination sensor, not a "slope circle".

The paper lacks in specific References (in relation to the landslide monitoring systems and related issues) and in an essential "discussion". The finding "the main cause ol collapse and slips appears to be the water content of soil" (see "conclusion") is previously revealed in "study area", so it seems that the manuscript does not make any original contributions.

I may suggest to the authors to work on the management focus "All the slopes were found to remain in a stable condition and the set threshold of vigilance values was not exceeded" (page 26), describing appropriately their case study with the purpose (as an example) to infer "general basics".

Reviewer 2 Report

Dear authors,

I tried to read your paper, but it was a hard task.

It is very confusing, english is poor, figures are very low quality, several section are useless (do you really believe that 3 pages about GPS are needed?), as well as figures from 9 to 16.

fig 18 is in Chinese, figg. 19 to 21 do not have the legends and y axis label

The introduction section do not provide any literature review, you mentioned a lot of places, but there is not any figure to understand where they are.

Figure contains several Chinese (I supposed Chinese) words and several mistakes.

some results are reported in the methodology section, section numbers are not ordered.

But the main problem is that it looks like you do not know what you are writing: what is a side slope disaster? what you mean with disaster monitoring?

You used fig. 30 to show the correlation between rainfall and displacements, but there is non any correlation in that figure.

what is fig. 31 for? To demonstrate that the lenses of the cameras are wet when it is raining?

find more comments in the attached file

Reviewer 3 Report

This manuscript is very interesting, well written in general. However, I have some minor concerns after carefully reading this manuscript.

Abstract:

- Abstract doesn’t have detail of method and main results. Therefore, it should add advantage and interesting details.

Introduction:

- Lecture of this research is too poor. Certainly, there are several similar your study in word. Please find and add those references.

Study Area:

- Maps of study area don’t have coordinate and clear legend. Please redesign those maps.

Historical Disasters:

- Important point of each earthquake is peak ground acceleration (PGA). Please describe earthquake in 1999. Also, describe about level of rainfall in 22 May 2007.

- Please add scale bar in Figure 3.

Side Slope Stability Simulation:

- I don’t find A-A’, B-B’, and C-C’ in map. Please change Fig 6. Also, legends aren’t clear.

Establishment of Collapse Monitoring Systems:

- How do you determine optimize position of GPS, geophones and etc.?

Introduction to GPS Measurement System Equipment:

- I think that you should combine “Introduction to GPS Measurement System Equipment”, “Single-Band GMX901 GPS Receiver” and “Spider Reference Station Real-Time Solution Network Software”.

Establishment of Management Values:

please write some of those references in introduction as lecture.

Other parts are so well-written.